# Possible Prevention of Diabetes with a Gluten-Free Diet

**DOI:** 10.3390/nu10111746

**Published:** 2018-11-13

**Authors:** Martin Haupt-Jorgensen, Laurits J. Holm, Knud Josefsen, Karsten Buschard

**Affiliations:** The Bartholin Institute, Ole Maaløes Vej 5, Rigshospitalet, 2200 Copenhagen, Denmark; laurits.juulskov.holm@regionh.dk (L.J.H.); knud@eln.dk (K.J.); buschard@dadlnet.dk (K.B.)

**Keywords:** beta cell, beta-cell stress, celiac disease, gluten-free diet, high-fat diet-induced obesity, intestinal permeability, islet of Langerhans, NOD mouse, type 1 diabetes, type 2 diabetes

## Abstract

Gluten seems a potentially important determinant in type 1 diabetes (T1D) and type 2 diabetes (T2D). Intake of gluten, a major component of wheat, rye, and barley, affects the microbiota and increases the intestinal permeability. Moreover, studies have demonstrated that gluten peptides, after crossing the intestinal barrier, lead to a more inflammatory milieu. Gluten peptides enter the pancreas where they affect the morphology and might induce beta-cell stress by enhancing glucose- and palmitate-stimulated insulin secretion. Interestingly, animal studies and a human study have demonstrated that a gluten-free (GF) diet during pregnancy reduces the risk of T1D. Evidence regarding the role of a GF diet in T2D is less clear. Some studies have linked intake of a GF diet to reduced obesity and T2D and suggested a role in reducing leptin- and insulin-resistance and increasing beta-cell volume. The current knowledge indicates that gluten, among many environmental factors, may be an aetiopathogenic factors for development of T1D and T2D. However, human intervention trials are needed to confirm this and the proposed mechanisms.

## 1. Gluten

During the recent years, there has been a tremendous increase in the number of GF products available with the promise of diverse health benefits. The incidence of celiac disease (CD) was estimated to be 33.6 per 10,000 person-years in a recent retrospective cohort study from the United Kingdom [1]. Non-celiac gluten sensitivity is thought to be more common, although the precise number is unknown [2,3,4,5]. Consumption of gluten is believed to affect many aspects of human health and is hypothesised to contribute to the diabetes pandemic, in which the number of people suffering from diabetes have quadrupled since 1980 to an estimated 422 million in 2014 [6]. Gluten entered our diet about 10,000 years ago in Mesopotamia when our ancestors began eating cereals. Today cereals are an essential food source around the world and more than 50% of the worlds daily caloric intake is derived from consumption of cereals [7]. 

Wheat, rye, and barley contain high amounts of gluten in their endosperm storage tissue. Chemically, gluten is classified as a prolamin, containing monomeric gliadins and polymeric glutenins. Glutenins can be subdivided into low and high molecular weight proteins while gliadins are divided into α-, γ-, and ω-gliadins [8]. The major amino acid constituents of gluten are proline, glutamine, and hydrophobic amino acids [9], which make gluten resistant to complete degradation by gastric, pancreatic, and brush-border enzymes [10,11]. α-gliadin contains some of the most toxic peptides in gluten, as evidenced by in vitro studies, and their effect has been mapped to specific domains in the structure [12]. The effect of the peptides are diverse and include gut-permeating [13], cytokine-releasing [14], and cytotoxic [15] effects. The 33-mer from α-gliadin is the most immunogenic gluten peptide discovered so far, as it contains three overlapping T cell epitopes [16], and processing by APCs is not required before T cell presentation [17]. The 33-mer is resistant to degradation by intestinal peptidases [18,19] and was recently identified in all of the modern and old cultivars of wheat and spelt analysed [20]. 

## 2. Gluten and T1D

T1D is initiated when autoreactive T cells destroy the insulin-producing beta cells in the pancreas leading to hypoinsulinaemia, and hyperglycaemia; however, the aetiology and pathogenesis are still not fully understood. T1D is classified as a multifactorial disease in which the genetic background, as well as environmental factors, are important determinants. Approximately 50% of the genetic risk of the disease is explained by the HLA class II region [21], and the haplotypes HLA-DR3-DQ2 and HLA-DR4-DQ8 are the most important individual genetic risk factors known [22].

In the period from 1990–1999, the average annual increase in the incidence of T1D was 2.8% in children aged ≤14 years [23]. Another epidemiological study predicted that the incidence would double between 2005 and 2020 in European children below five years of age [24]. 

A key question is why the incidence of T1D is increasing now when gluten was introduced 10,000 years ago. Among many possible reasons, such as an increased exposure to other diabetogenic environmental factors, a recent study in non-obese diabetic (NOD) mice, an animal model of autoimmune diabetes, found that modern wheat sources are more diabetogenic that old wheat sources [25].

### 2.1. The Role of Environmental Factors

Environmental factors are important in the pathogenesis of T1D. First, the incidence of T1D has been rising globally at a pace that cannot be explained by genetic drift [23,24]. Second, increasing incidence has been observed in population groups that have migrated from regions with low incidence of T1D to regions with high incidence [26]. Third, a six-fold gradient in the incidence of T1D is observed between Russian Karelia and Finland, although the frequency of the high-risk HLA-DQ genotypes is equal in the two populations [27], and similar gradients between neighbouring countries are also known [23]. Fourth, monozygotic twins are most often discordant for T1D [28,29]. Fifth, T1D develops in less than 10% of subjects with HLA-conferred risk genotypes [30], although all risk genotypes are most likely not identified yet. 

Many environmental factors have been associated with increased susceptibility to T1D, including physiological stress, vaccines, toxins, cow milk [31], and dietary gluten. Evidence for a viral aetiology has grown during the recent years [32,33,34,35] exemplified by the Diabetes virus detection (DiViD) study that demonstrated low-grade enterovirus infection in islets from the majority of the newly diagnosed T1D patients investigated but not in any of the non-diabetic controls [36]. In this regard, the hygiene hypothesis is central, stating that children who are exposed to microorganisms will develop strong immunity against these, which will dampen the harmful effects from them, but also protect the child from T1D [37].

Thus, T1D is a multifactorial disease where many environmental factors are likely to contribute to the pathogenesis, including gluten.

### 2.2. GF Diet, Early Evidence and Timing

Early studies in NOD mice [38] and biobreeding (BB) rats [39], which are animal models for autoimmune diabetes, suggested that cereals might have a role in the aetiopathogenesis of T1D. Later, a study in NOD mice demonstrated that a lifelong GF diet compared to a gluten-containing standard (STD) diet reduced the incidence of autoimmune diabetes from 64% to 15%, although insulitis score was not significantly reduced [40], and subsequent studies in NOD mice demonstrated similar results [41,42]. In a more recent study in NOD mice, we showed that the incidence of autoimmune diabetes could be reduced even further, to 8%, together with reduced insulitis in offspring, by keeping the mothers on a GF diet exclusively during pregnancy [43]. Similar, but smaller effects on incidence and insulitis in NOD mouse offspring were demonstrated by keeping the mice GF in utero and in early postnatal life [44]. The GF diets used it these studies had the gluten protein replaced with other proteins (meat, casein, or egg white) while keeping an equal content relative to the STD diets of protein, fat, and components that may influence the risk of diabetes such as milk and soybean [40,43]. Although the major difference between the GF and STD diets was gluten, small differences were present between other dietary components. Human evidence for the existence of an early time-window for the introduction of gluten and tolerance induction includes the BABYDIAB and Diabetes and Autoimmunity Study in the Young (DAISY) cohort studies. These studies showed that the risk of islet autoimmunity increased if gluten was introduced before the age of three months compared with receiving only breast milk during this period [45] or first exposure to gluten between age four and six months [46], after adjustment for covariates. Moreover, it was shown that breastfeeding during the introduction of gluten was associated with decreased risk of islet autoimmunity in children at high risk of T1D [46]. Recently, we published a study based on the Danish National Birth Cohort, which demonstrated that maternal ingestion of low versus high amounts of gluten during pregnancy reduced the risk (2-fold) of T1D in their children, after adjustment of covariates [47]. Other cohort studies showed no association between intake of GF diet during pregnancy and T1D, again after adjustment of covariates [48,49]. In two Danish studies, a GF diet was administered to children after T1D diagnosis and the children showed improvements in disease parameters including prolonged partial remission periods and reduced HBA1c compared to control children with T1D matched by diabetes duration and age [50,51]. Another study investigated how children at increased risk of T1D responded to six months of GF diet followed by six months of gluten-containing diet (diets were not matched for carbohydrate content etc.) [52]. Following the GF diet, the children showed improved glucose tolerance and insulin sensitivity (non-significant) but unchanged titres of islet autoantibodies. Following the months of the gluten-containing diet, the study reported a decreased insulin sensitivity. Hence, a GF diet may have a preserving effect on beta-cell function on older children with T1D.

In summary, the studies suggest that a GF diet may have the potential to reduce the risk of T1D. Interestingly, a GF diet seems to be most effective when applied in utero, and timing of the introduction of gluten is apparently critical. Moreover, a few studies indicate that a GF diet, when applied to older children with T1D, may preserve the beta cells to some extent.

### 2.3. GF Diet and the Intestine

The intestinal microbiota seems to play an important role in the pathogenesis of T1D but causality is still unclear. Patients with T1D have an increased intestinal permeability [53,54,55,56] and show a decrease in bacteria that maintain the intestinal permeability [57,58]. Perturbation of the intestinal microbiota in childhood is thought to disturb the developing immune system and may thus be a pathogenic factor [59,60]. NOD mice fed a GF versus a STD diet had overall fewer bacteria as well as fewer aerobic and microaerobic bacteria in caecum [41]. In a similar study, NOD mice on a GF diet showed decreased *Bifidobacterium*, *Tanerella* and *Barnesiella* species and increased *Akkermansia* species in faeces [42]. A GF diet during pregnancy and early postnatal life has been demonstrated to induce pronounced differences in the intestinal microbiota of NOD mouse mothers and offspring, including increased numbers of bacteria from the phylae *Akkermansia*, *Proteobacteria*, and *TM7* [44]. The mucin-degrading *Akkermansia* is of special interest in T1D. For example, NOD mice treated with vancomycin from an early age had increased proportions of *Akkermansia* and reduced incidence of autoimmune diabetes [61]. In addition to the association to T1D, *Akkermansia* species reversed the increased intestinal permeability in Apolipoprotein E (Apoe)−/− mice and decreased the entry of lipopolysaccharide (LPS) into the circulation [62]. Interestingly, a study in children from the BABYDIET study showed that *Bacteroides*-dominated children were more likely to develop islet autoantibodies and had decreased potential to butyrate production compared to *Akkermansia*-dominated children [63]. Short-chain fatty acids (SCFAs) are produced by bacteria during breakdown of dietary fibre and include butyrate, acetate, and propionate. Butyrate and acetate diminish the intestinal permeability [64,65], and butyrate can boost the number and function of regulatory T cells (Tregs) [65,66], which are known to suppress inflammatory responses. Acetate can reduce the proportion of autoreactive T cells [65]. 

Thus, a GF diet may improve the intestinal microbiota and permeability (Figure 1), but more studies are needed in order to gain knowledge about mechanisms and causality.

### 2.4. GF Ddiet and the Immune System

We and others have conducted a range of animal studies, which suggest that a GF diet modulates the innate and adaptive immune system (Table 1).

A GF diet reduced the natural killer (NK) cell activity in pancreatic lymph nodes (PLNs) from Bagg albino (BALB/c) mice and in spleen from NOD mice compared to mice on a STD diet [68]. Another study in NOD mice confirmed this observation in spleen and found reduced activity of cytotoxic T lymphocytes (CTLs) in PLN [69]. Besides NK cells, macrophages from mice have been shown to produce proinflammatory cytokines (interleukin (IL)6, IL12 and tumor necrosis factor alpha (TNFA), among others) upon gliadin stimulation [74]. Dendritic cells (DCs) may also be affected by a GF diet, as the diet reduced the proportions in thePLN and mesenteric lymph node (MLN) and increased the proportions of tolerogenic DCs in PLN in BALB/c mice [69]. This study also found downregulation of DC activation markers in lymphoid organs from the GF mice. This is supported by a study showing that gliadin stimulation of bone marrow-derived DCs from BALB/c mice resulted in maturation of the cells, as seen by evaluation of activation markers and chemokines (keratinocyte-derived cytokine (KC) and macrophage inflammatory protein 2 (MIP-2)) [75]. Moreover, gliadin stimulation may increase the expression of toll-like receptor (TLR)4, 7, 8 and interferon alpha (IFNA) in DCs of DQ8 transgenic mice [76]. As for the adaptive immune system, we have shown that a GF diet reduced the proportion of T helper (TH)17 cells in PLN of BALB/c mice [72] and dampened the inflammatory profile of T cells [73]. A wheat-free versus a wheat-based diet has been shown to reduce the proportion of TH1 cells in MLN from young diabetes-prone BB rats [71]. Three studies in NOD mice found that a GF diet during pregnancy alleviated T1D in the offspring and that the mechanisms were likely to involve changes in the immune system. The first study showed that the intestines from the offspring exposed to the GF diet in utero had an increased gene expression of immunosuppressive M2 macrophages and reduced expression of proinflammatory cytokines [44]. This study also showed that the GF diet increased the proportion of Tregs in PLN and decreased the proportion of DCs in PLN, MLN, and spleen in the offspring. Moreover, the traffic of T cells between PLN and the gut-associated lymphoid tissue (GALT) was increased. Recently, we demonstrated that a GF diet during pregnancy reduced the expression of interferon gamma (IFNG) in TH cells and IL22 in gamma delta T receptor (gdTCR)^+^ T cells from spleen of NOD mouse offspring [67]. This is of interest because IFNG [77] and IL22 [78] might have a role in T1D. Furthermore, we demonstrated a reduced inflammatory profile in subpopulations of T cells from lymphoid organs of offspring exposed to a GF diet in utero [67]. In another study, we observed that the gene expression of TH17 cells was reduced in colon from NOD mice exposed to a GF diet in utero [43]. These results were confirmed in NOD mice fed an anti-diabetogenic diet compared to a wheat-based diet from a young age [70]. The mice showed reduced numbers of TH17 cells, besides reduced numbers of activated TH cells and DCs and reduced incidence of autoimmune diabetes. Taken together, a GF diet has a dampening effect on the innate and adaptive immune system, as evidenced in different animal models (Table 1).

Human evidence of the effect of gluten on the immune system is more limited. Early evidence includes a study from 1987 demonstrating increased titres of anti-gliadin antibodies at the onset of T1D in 54% of patients with no signs of CD and in none of the healthy controls [79]. Moreover, a study in children with T1D and no CD showed that rectal gluten application resulted in an increased percentage of epithelium and lamina propria CD3^+^ and gdTCR^+^ cells in 26% of the patients compared to approximately 15% of the healthy controls [80], indicating that some T1D patients have an abnormal mucosal immune response towards gluten. In another study, 47% of the included patients with T1D and no CD and none of the healthy controls displayed increased proliferation of peripheral blood mononuclear cells (PBMCs) after stimulation with wheat polypeptides, and the cytokine response was a proinflammatory TH1/TH17 [81]. On the other hand, CD4^+^ cells from children with pre-T1D and multiple islet autoantibodies compared to healthy controls showed decreased proliferative responses after gliadin stimulation [82]. DCs may be affected by gluten as well. This is shown in a study where gliadin stimulation of monocyte-derived DCs from healthy donors resulted in increased expression of maturation markers (CD80, CD83, CD86, and HLA-DR molecules), increased production of chemokines and cytokines (TNFA, IL6, IL8, IL10 among others), increased capacity to stimulate proliferation of allogeneic T cells, as well as reduced endocytic activity [83].

Thus, the studies suggest that a GF diet reduces inflammation in the intestines and pancreas, involving many cell types of the immune system (Figure 1).

### 2.5. GF Diet, Risk of T1D and CD

CD is a chronic autoimmune disease that results in inflammation of the intestinal submucosa and destruction of the intestinal epithelium. The following clinical signs are typically observed: increased numbers of intraepithelial lymphocytes, villus atrophy, and crypt hyperplasia. The treatment consists of a strict GF diet and if not initiated it will lead to undernourishment. The pathogenesis is thought to start with binding of gliadin to the chemokine receptor CXCR3 on enterocytes, which results in increased intestinal permeability by myeloid differentiation primary response 88 (MyD88)-dependent zonulin release and crossing of the lamina propria by gliadin [13]. At this site, tissue transglutaminase (tTG) deamidates glutamine residues in gliadin to glutamate, which mediates high affinity of gliadin to HLA-DQ2/DQ8 on APCs and thus activation of CD4^+^ T cells specific for gliadin [84,85]. Next, production of proinflammatory cytokines and activation of CD8^+^ T cells specific for gliadin are thought to further worsen the damage to the intestine [86]. 

Interestingly, human studies indicate that T1D and CD are comorbid diseases and they share the genetically predisposing haplotypes HLA-DQ2/DQ8. The risk of developing other autoimmune diseases is increased for patients with CD, in particular for those diagnosed early (<16 years of age [87]) (<18 years of age [1]). The prevalence of CD is 5–10% in patients with T1D [88,89,90,91,92]; however, the risk is lower in patients compliant to a GF diet [87]. The majority of NOD mice [93] and 12% of patients with T1D are seropositive for anti-tTG [94]. In addition, NOD mice on a STD versus a GF diet have reduced villus height and increased infiltration of intraepithelial lymphocytes (enteropathy) [95], and human subjects exhibit signs of enteropathy already from the pre-diabetic stage [54]. Jejunal biopsies from children with T1D versus healthy controls were stimulated with gliadin [96]. The result was increased proportions in lamina propria of intraepithelial T cells and increased expression of CD25^+^ (IL2RA), CD80^+^ (co-stimulatory), intracellular adhesion molecule 1 (ICAM-1), and crypt HLA-DR. Thus, NOD mice and patients with T1D exhibit signs of CD, which underlines that the two diseases are associated and indicates that gluten may be a common environmental factor. This is supported by a study in NOD mice, which showed that a GF versus a STD diet decreased intraepithelial infiltration of T cells, expression of IFNG, enteropathy and incidence of autoimmune diabetes [95]. Moreover, we recently published a study showing that young NOD mice exposed to a GF versus a STD diet in utero had persistently reduced titres of anti-tTG in serum together with increased villus-to-crypt (V:C) ratios (improved enteropathy) [67], indicating that a GF diet during pregnancy may not only ameliorate T1D but also dampen the symptoms of CD in the offspring. 

Hence, studies in animals and humans with T1D indicate that a GF diet may reduce the signs of CD and that the prenatal period may be of special importance (Figure 1).

### 2.6. Mucosal Tolerance Induction by Gliadin

A study showed that BB rats fed wheat gluten neonatally versus at weaning were protected from autoimmune diabetes [97]. It was later demonstrated that NOD mice exposed to a diet with a three times higher content of gluten compared to a normal diet in utero and the rest of their lives were protected from autoimmune diabetes to the same extent as those on a GF diet [98]. The authors proposed that the high amount of gluten might result in mucosal tolerance or unresponsiveness, which was also seen in a human monocyte cell line regarding LPS in high doses and long exposure times [99]. More recently, administration of gliadin intranasally to four-week-old NOD mice was shown to decrease the incidence of autoimmune diabetes and insulitis and increase the numbers of gdTCR^+^ cells and forkhead box P3 (FOXP3)^+^CD4^+^ Tregs in mucosal lymphoid organs [100]. Intestinal gdTCR^+^ cells are important for the induction of peripheral Tregs and during induction of mucosal tolerance they seem to have a central role in maintenance of tolerance [101]. Tregs are functionally deficient in patients with T1D [102]. The hypothesis on mucosal tolerance induction against gluten is backed by an exploratory registry-based case-control study that we conducted, which demonstrated that occupation with grain crops, i.e., by bakers, was associated with lower incidence of T1D [103]. We speculated that the lower incidence among workers occupied with grain crops was due to nasal mucosal exposure to gluten during work and hence tolerance induction. 

These studies show that a possible strategy in the prevention of T1D may be induction of mucosal tolerance against gluten, but human intervention studies are needed to confirm this.

### 2.7. GF Diet and the Beta Cell

From studies in NOD, C57BL/6 (B6), and BALB/c mice, we demonstrated that gliadin peptides cross the intestinal barrier after oral gavage and thereafter localise to the pancreas and to a smaller extent the islets [104]. The ability of gluten peptides to cross the intestinal barrier has been independently confirmed [105]. Gliadin does also seem to cross the intestinal barrier in humans, evidenced by observations of gliadin in breast milk and serum from healthy mothers [106]. Gluten peptides are likely in close contact with the beta cells because of the high degree of vascularisation in islets [107]. In vitro studies on INS-1E insulinoma cells and isolated rat islets showed that gliadin increased glucose-stimulated insulin-secretion (GSIS) by closing ATP-dependent K-channels [108]. This could be part of the pathogenesis of T1D by means of gliadin-mediated beta-cell hyperactivity and thus increased expression of islet antigens and autoimmunity [109]. A gluten-containing STD versus a GF diet is known to increase insulitis in animal models of T1D [39,40,43,44,67], and inflammatory cell-stress increases the expression and enzyme activity of tTG [110]. tTG has been shown to induce posttranslational modifications of human islet antigens and thereby increase the affinity to HLA-DQ [111]. Thus, it is likely that a GF diet reduces the tTG activity in islets and reduce insulitis. Interestingly, NOD mice exposed to a GF diet exclusively in utero had increased numbers of islets throughout the prediabetic phase, besides reduced insulitis and autoimmune diabetes incidence [67]. The finding is in agreement with an older study in BB rats, which were fed a diet based on hydrolysed casein versus a diet based on cereals from early life [39]. The rats had increased total islet area and numbers, together with lower insulitis and autoimmune diabetes incidence.

Altogether, it is likely that a GF diet reduces beta-cell stress and this may result in increased numbers of islets, besides reduced insulitis and autoimmune diabetes incidence, an effect that has also been observed when the diet was applied in utero (Figure 1).

## 3. Gluten and T2D

T2D is associated with obesity and the incidence is expected to increase between 2010 and 2030 [112]. Overall, T2D is a result of insulin resistance and beta-cell dysfunction [113]. Although insulin resistance is often present in obese subjects, their beta cells initially compensate by increasing the insulin production [114] and mass [115]. Eventually beta-cell dysfunction occurs, resulting in hyperglycaemia and diabetes [116]. As for T1D and CD, genetic susceptibility genes are important disease determinants in T2D, and so far studies have found over 40 associated genes, although only a few of them have been verified in several patients and laboratories including peroxisome proliferator activated receptor gamma (*PPARG*), ATP binding cassette subfamily C member 8 (*ABCC8*), and potassium voltage-gated channel subfamily J member 11 (*KCNJ11*) [117].

### 3.1. The Role of Environmental Factors

The rising incidence [6,112] and the observation that monozygotic twins are often discordant for T2D [29,118,119] indicates that environmental factors, besides genetic susceptibility genes, are important for the development of T2D. Obesity, a result of imbalance between energy intake and expenditure because of excess intake of food and insufficient physical activity [120], most likely plays a causal role in the pathogenesis of T2D [121]. According to the thrifty gene hypothesis, genes that were advantageous for accumulation of adipose tissue during times of caloric excess in the previous hunter/gatherer period might explain the present rise in the incidence of T2D in populations with caloric excess [122]. The intrauterine environment may be of special importance, in which pesticides [123], hormonal agents, patterns of feeding and undernutrition are potential determinants [120]. In this context, the thrifty phenotype hypothesis should be mentioned. It suggests that poor foetal and early postnatal nutrition leads to insulin resistance and beta-cell dysfunction in the adult, which, in combination with the effects from ageing and obesity, manifests in T2D [124].

### 3.2. GF Diet, Leptin Resistance and the Link to Obesity

Leptin resistance is likely involved in the pathogenesis of obesity and thus T2D [125,126]. Leptin resistance has been hypothesised to be the result of insufficient genetic adaptation to a cereal-based diet [127], as humans began consuming cereals only 10,000 years ago. The hypothesis was tested in piglets receiving a palaeolithic diet, i.e., a diet containing nuts, vegetables etc., versus a cereal-based diet from the age of 2–17 months [128]. Following this period, the palaeolithic piglets showed reduced body weight, subcutaneous fat thickness, and pancreatic lymphocyte numbers but increased insulin sensitivity. Interestingly, clinically relevant concentrations of trypsin- and pepsin-digested wheat gluten was demonstrated to hinder the binding between leptin and its receptor, indicating that gluten could be linked to leptin resistance and obesity [125].

### 3.3. GF Diet and the Intestine

Obesity and T2D are associated with intestinal dysbiosis [129]. Obese subjects have changes in the intestinal microbiota (↑phylum *Firmicutes* and ↓phylum *Bacteroides*) [130], which, as seen in mice, increase the capacity of the microbiota to harvest energy and increase fat stores [131]. A large study in T2D patients showed a reduced abundance of bacteria (*Roseburia*, among others) capable of producing the SCFA butyrate [132]. SCFAs have been demonstrated to prevent high-fat (HF) diet-induced obesity in B6 mice [133] and studies in Caco-2 epithelial cells showed that butyrate may decrease intestinal permeability [64], which is increased in T2D patients [134,135]. Interestingly, gliadin stimulation of intestinal tissue from mice and humans increased the permeability [13], and studies in different mouse strains found that oral gavage of gluten peptides resulted in accumulation in extraintestinal organs [104,105]. Regarding the intestinal permeability, B6 mice fed a gluten-free high-fat (GF-HF) diet versus a gluten-containing HF diet showed improved intestinal barrier function [136]. Moreover, the GF-HF mice showed changes in the intestinal microbiota (↑genus *Lactobacillus* and ↓genera *Clostridium XI*, *Coriobacteriaceae* and *Dorea*) associated with improved health [137,138,139,140]. Moreover, the abundance of *Akkermansia* species was reduced in GF-HF versus HF mice, which is puzzling for several reasons. First, increased intestinal permeability and leakage of LPS to the circulation may be reversed by the bacterial species *Akkermansia muciniphila*, as shown in Apoe−/− mice [62]. Second, B6 mice receiving oral gavage of the bacterium showed reduced HF diet-induced metabolic disorders i.e., metabolic endotoxaemia, adipose tissue inflammation, insulin resistance, and fat mass gain [141]. Third, *Akkermansia muciniphila* is found in lower abundance in intestines of pre-T2D patients compared with healthy controls [142].

Studies in mice show that the early onset of HF diet-induced hyperglycaemia is associated with increased leakage of LPS and gram-negative bacteria from the intestine to the adipose tissue, which is thought to continuously fuel metabolic bacteraemia and endotoxaemia [143], and may contribute to low-grade inflammation, insulin resistance, beta-cell dysfunction, and, thus, T2D. This is relevant because intake of gluten seems to both increase the intestinal permeability and lead to a disease-associated intestinal microbiota. The intake of gluten could therefore contribute to T2D by the above-mentioned mechanism.

Thus, studies in mouse models of T2D indicate that a GF diet may improve the intestinal barrier function and lead to a healthier microbiota, both of which could alleviate the disease by reducing passage of inflammatory gluten peptides, bacterial products etc. (Figure 2).

### 3.4. GF Diet, the Beta Cell and Adipose Tissue

Signalling through the pattern-recognition receptor TLR4, which is expressed on most cell types [144], has been associated with insulin resistance [145] and beta-cell dysfunction in T2D [146]. This is relevant because gliadin may activate signalling through this receptor [147]. Thus, gliadin may induce insulin resistance and beta-cell dysfunction through TLR4 and possibly also through other innate immune receptors. Moreover, we have demonstrated that gliadin peptides affect the beta cells directly, as the peptides increase GSIS in rat islets and INS-1E cells and potentiate the fatty acid-stimulated insulin secretion in INS-1E cells [108].

These studies imply that a GF diet may alleviate insulin resistance as well as beta-cell stress and dysfunction in T2D (Figure 2). However, the results need further confirmation.

### 3.5. GF Diet, the Immune System, Obesity and T2D

We have shown that intravenous injections with enzymatically degraded gluten increase the body weight of NOD mice [108]. Likewise, rats fed a casein-based versus a gluten-based diet for two weeks had reduced liver lipogenesis [148]. Moreover, B6 mice fed a GF-HF versus a HF diet for eight weeks had reduced concentrations in serum of the proinflammatory adipokines leptin and resistin and increased concentrations of the anti-inflammatory adipokine adiponectin, together with reduced body weight, epididymal fat stores, fasting glucose, and insulin [149]. This indicated that gluten could directly contribute to obesity and hence T2D. The improved glucose homeostasis in the GF-HF mice was thought to involve a reduced inflammatory profile leading to increased expression of *PPARG*, an important regulator of lipid metabolism, and thus increased expression of adiponectin and glucose transporter 4 (*GLUT-4*) i.e., improved insulin sensitivity [149]. Similar observations were seen in another study in B6 mice fed a GF diet [105], both in normal and HF settings. The study also showed that increased thermogenesis and energy expenditure were behind the observed effects. We showed that long-term feeding of B6 mice with a GF-HF versus a HF diet increased the beta-cell volume and improved the glucose tolerance [150], which we believe could be a result of beta-cell rest, as we have shown that gluten potentiates the fatty acid-stimulated insulin secretion [108]. Not all animal studies have been able to demonstrate these anti-obesity and anti-diabetes effects from a GF diet. As an example, Apo−/− mice were fed a GF diet in utero and until 16 weeks of age, but no effect was observed on body weight, glucose tolerance, insulin levels, and plasma lipids; however, a transient change was seen in the intestinal microbiota [151]. Thus, the effects of a GF diet on obesity and T2D have been tested only in a few animal studies, some of which indicate that a GF diet may have the potential to reduce obesity and T2D (Figure 2).

In a randomised, crossover study, T2D patients were fasted and received a test meal differing in the protein sources whey, casein, cod, and gluten [152]. After eight hours, the patients from the gluten-group showed increased incremental area under the curve (AUC) for plasma glucose compared to patients from the other groups. Moreover, the patients eating gluten showed increased incremental AUC for fatty acids and triglycerides compared to patients receiving whey. The national health and nutrition examination survey (NHANES) study, which consists of a number of cross-sectional surveys carried out every second year, demonstrated higher high-density lipoprotein (HDL), smaller waist circumference and self-reported weight loss in persons on a GF diet [153]. Further human evidence includes a study in children with autism spectrum disorder on a GF casein-free diet versus a regular diet for three months, showing lower body weight and body mass index [154]. A recent prospective cohort study from the USA based on the Nurses’ Health Study I and II and the Health Professionals Follow-Up Study found an inverse association between intake of gluten and risk of T2D [155]. The analyses were adjusted for relevant covariates such as fibre and folic acid, which are often low in a GF diet, and the association was slightly weakened after adjustment of cereal fibre. Excluding gluten from the diet will mediate exclusion of other potential anti-diabetogenic factors; hence, this study is not conclusive and should be followed up with an intervention study. In summary, a few intervention studies have shown that a GF diet may alleviate obesity and T2D in humans (Figure 2). On the other hand, a recent prospective cohort study report that a GF diet is associated with T2D. Thus, larger intervention studies clarifying the effect of a GF diet on T2D patients are needed. 

## 4. Conclusions

In this review, we looked at the role of dietary gluten as a contributing factor in the aetiopathogenesis of T1D and T2D. Gluten has multiple effects in the body starting in the intestine where it affects the composition of the microbiota, may induce enteropathy in T1D, and increases the intestinal permeability, all of which seem to improve with a GF diet. In animals, gluten has been found to cross the intestinal barrier together with LPS among other substances and may accumulate in different tissues including islets and adipose tissue. In animal beta cells, gluten peptides induce insulin secretion, an effect that is potentiated by palmitate, suggesting that gluten peptides may induce beta cell-stress, -dysfunction, -loss, and autoimmunity, and thus contribute to both T1D and T2D. Moreover, gluten peptides may also contribute to leptin- and insulin resistance, regarding obesity and T2D. Primarily animal studies have shown that a GF diet dampens the innate and adaptive immune system leading to a less inflammatory profile. In T1D, the timing of gluten introduction to children is most likely critical, and the efficiency of gluten exclusion during pregnancy, combined with early introduction of gluten postnatally, should be investigated as a prevention strategy in an intervention study. Interestingly, animal studies suggest that a GF diet in utero may also reduce the risk of CD and report changes in the pancreas morphology including increased numbers of islets. Postnatally, a GF diet remains an interesting therapeutic option for the prevention and treatment of T1D, but more human intervention studies must be carried out. Mucosal tolerance induction of gluten is another potential strategy in reducing the risk of T1D but this also needs further exploration.

In T2D, the evidence for an alleviating effect of a GF diet is more uncertain especially regarding the few human studies that have been conducted, although animal studies report improvements of both obesity and T2D. A GF diet is typically low in fibres and other antidiabetogenic nutrients, and studies investigating the long-term effects on obesity and T2D of gluten specifically are therefore needed.

## Figures and Tables

**Figure 1 nutrients-10-01746-f001:**
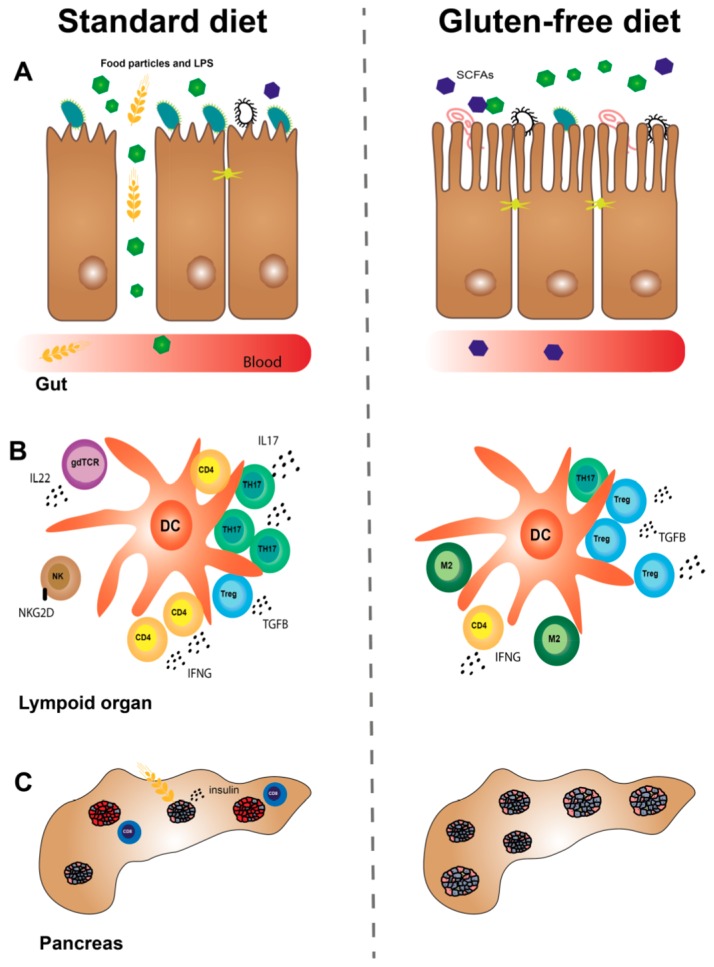
Gluten free (GF) diet and the development of type 1 diabetes (T1D)—a hypothesis. (**A**) A GF diet decreases the intestinal permeability and increases the villus-to-crypt (V:C) ratio, thereby preventing food particles such as gliadin peptides from crossing the intestinal barrier and reacting the pancreas. A GF diet increases the number of *Akkermansia* bacteria, among other changes, and the amount of short-chain fatty acids (SCFAs) such as butyrate. (**B**) A GF diet modulates the innate and adaptive immune system resulting in reduced interferon gamma (IFNG) secretion from CD4^+^ T helper (TH) cells, reduced interleukin (IL)22 secretion from gamma delta T cell receptor (gdTCR)^+^ T cells, and lower numbers of activated (NKG2D^+^) natural killer (NK) cells, among other things. TH17 cell numbers are reduced and immunosuppressant M2 macrophage numbers and forkhead box P3 (FOXP3)^+^ regulatory T cell (Treg) numbers are increased. (**C**) A GF diet reduces beta-cell stress by reducing the insulin secretion. This may preserve the number of islets, reduce insulitis, and ameliorate T1D.

**Figure 2 nutrients-10-01746-f002:**
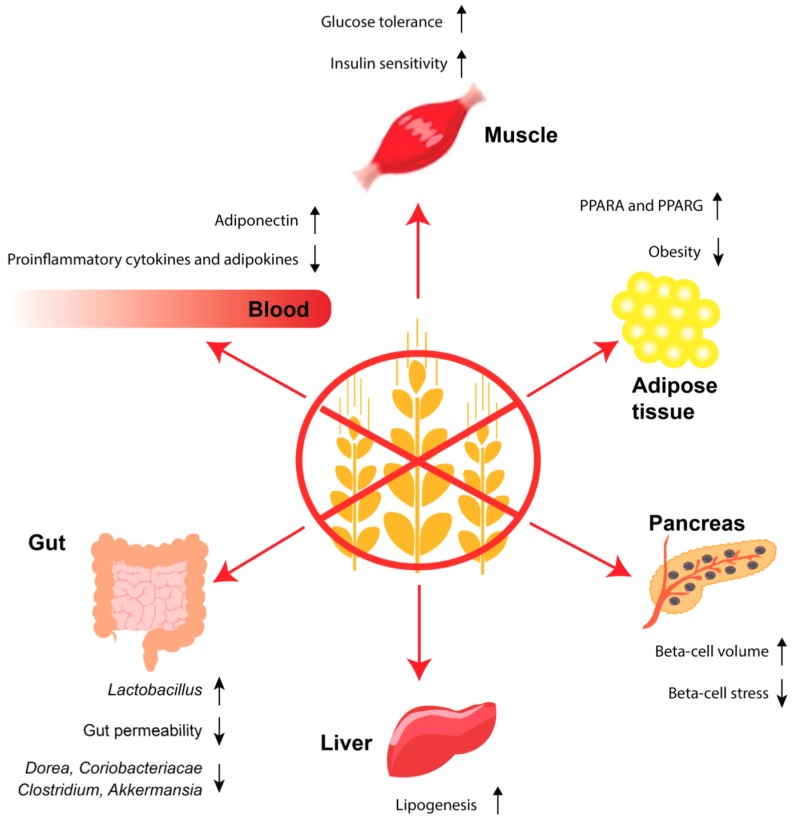
Gluten-free (GF) diet and the development of type 2 diabetes (T2D)—a hypothesis. A GF diet decreases intestinal permeability thereby preventing food particles such a gliadin from crossing the intestinal barrier and reaching the adipose tissue and pancreas. A GF diet increases the proportion of *Lactobacillus* and decreases the proportion of *Akkermansia*, *Dorea*, *Clostridium*, and *Coriobacteriacae*. In the blood, a GF diet decreases the level of proinflammatory cytokines and adipokines and increases the anti-inflammatory adiponectin. A GF diet reduces obesity and improves the regulation of lipid metabolism by upregulating peroxisome proliferator activator receptor alpha (PPARA) and peroxisome proliferator activator receptor gamma (PPARG) in adipose tissue. This, in turn, leads to increased insulin sensitivity and improved glucose tolerance, which is further improved by reduced beta-cell stress and increased beta-cell volume.

**Table 1 nutrients-10-01746-t001:** An overview of some of the effects that a gluten-free (GF) diet has on the immune system in animal models of type 1 diabetes (T1D).

**Immunological Effects of a GF versus a STD Diet in Utero in NOD Mice**	**References**
↑M2 macrophage gene expression in intestine.	[44]
↓DC (CD11b^+^CD11c^+^) numbers in PLN, MLN and spleen.	[44]
↓TH1 cell (IFNG^+^CD4^+^) numbers in spleen.	[67]
↓TH17 (*RORGT*) gene expression in colon.	[43]
↓gdTCR cell (IL22^+^gdTCR^+^) numbers in spleen.	[67]
↑Treg cell (FOXP3^+^CD4^+^) numbers in PLN.	[44]
↑T cell (α4β7^+^CD4^+^/CD8^+^) numbers in PLN.	[44]
↓proinflammatory cytokine gene expression in intestine.	[44]
↓T cells (gdTCR^+^, CD4^+^, CD8^+^ and FOXP3^+^) inflammatory cytokine profile in spleen, PLN, MLN and ILN.	[67]
**Immunological Effects of a GF versus a STD Diet Postnatally in Animal Models of T1D**	**References**
↓NK cell (activated) (NKG2D^+^NKp46^+^/DX5^+^) numbers in PLN (BALB/c mice) and spleen (NOD mice).	[68,69]
↓DC (CD11b^+^CD11c^+^) numbers in PLN and MLN (BALB/c mice).	[69]
↓DC (CD11b^+^CD11c^+^) numbers in colon (NOD mice).	[70]
↓DC (activated) (CD40^+^/CCR7^+^/MHCII^+^ on CD11c^+^) numbers in different lymphoid organs (BALB/c mice).	[69]
↑DC (tolerogenic) (CD103^+^CD11b^+^) numbers in PLN (BALB/c mice).	[69]
↓TH cell (CD4^+^) numbers in colon (NOD mice).	[70]
↓TH1 cell (IFNG^+^CD4^+^) numbers in MLN (BB rats).	[71]
↓TH17 cell (IL17^+^CD4^+^) numbers in PLN (BALB/c mice).	[72]
↓TH17 cell (IL17^+^CD4^+^) numbers in colon (NOD mice).	[70]
↓T cell (CD3^+^) proinflammatory profile (BALB/c mice).	[73]
↓CTL (activated) (NKG2D^+^CD8^+^) numbers in PLN (NOD mice).	[69]

↑: increased; ↓: decreased.

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
