# Peer review of "Possible Prevention of Diabetes with a Gluten-Free Diet"

_nutrients, 2018, doi:10.3390/nu10111746_

Round 1
Reviewer 1 Report
This is an excellent review of the possible role of gluten in T1D and T2D. The authors have drawn together appropriate support from the literature for their argument and have been circumspect in the interpretation of the evidence. I thought the authors could consider attempting to clarify and differentiate the effect of a gluten-free diet being due simply to the lack of gluten (i.e., an immune stimulant) versus indirect effects mediated via changes in permeability, gut microbiota, alterations in gut immunity and the target beta cells. They do attempt this but it might be more succinctly put. Otherwise, I found the MS to be well written and a useful piece of synthesis with a refreshing new hypothesis regarding a potential role for gluten in T2D.
Author Response
This is an excellent review of the possible role of gluten in T1D and T2D. The authors have drawn together appropriate support from the literature for their argument and have been circumspect in the interpretation of the evidence. I thought the authors could consider attempting to clarify and differentiate the effect of a gluten-free diet being due simply to the lack of gluten (i.e., an immune stimulant) versus indirect effects mediated via changes in permeability, gut microbiota, alterations in gut immunity and the target beta cells. They do attempt this but it might be more succinctly put. Otherwise, I found the MS to be well written and a useful piece of synthesis with a refreshing new hypothesis regarding a potential role for gluten in T2D.
Thank you for the comment. We agree that the relationship between intake of a gluten-free diet and changes in intestinal permeability, microbiota, immune system and beta cells is still not clear, although there are good animal studies out there and we and others are working on clarifying it from primarily studies in NOD mice. In an attempt to clarify and add depth to the manuscript on these matters, we have added more evidence especially on the effect on a gluten-free diet on the intestinal microbiota. We have also toned down the conclusions throughout the manuscript to avoid the risk of misleading the reader regarding causality. Moreover, we have added important details to the descriptions of the studies discussed in the manuscript so that the reader may draw their own conclusions more easily.
Reviewer 2 Report
The authors present a very well structured paper on the topic of gluten and diabetes risk. There are a few areas which would benefit from some clarifications. These have been outlined below.
1. If gluten entered our diet about 10,000 years ago and yet the annual increase in incidence of T1D has only been on the rise in the past few decades, it is unclear how gluten likely contributes to this rise in incidence. A bridge between this information and the mechanistic data discussed from animal models is required to help with the flow of the first part of the paper.
2. In the section 2.2, the conclusion seems a bit strong for the data presented. For example only 1/3 studies mentioned found an effect of a GF diet during pregnancy and risk of T1D in their children and reference 49 was not significant. It is important to be transparent, thus more descriptions of the studies should be added, including limitations of the evidence in order to promote an unbiased discussion. For example, regarding references 47 and 48, please describe if there was a control group. If not, the study may be confounded by post-diagnosis standard treatment. Also, please indicate what gluten is being replaced with in the GF diets as this can be relevant to any effect observed. The conclusion drawn in this section is also a principal statement in the abstract, thus may also require modification.
3. In section 2.3, again the conclusion does not seem well supported by the evidence presented. There was only one study mentioned which was in NOD mice which included a GF intervention. This section is lacking any discussion of the effect of a GF diet.
4.There is a general lack of human studies on gluten and T1D. There is some very limited evidence presented on effects when consumed during pregnancy, but none other than this. Furthermore, the case-control study in section 2.6 demonstrated a lower incidence in those with an occupation with grain crops, i.e. bakers. This further adds to the debate on any effect of a GF diet on T1D risk. The lack of information in human studies in T1D in order to draw any strong conclusions should be made clearer in the abstract. It is also suggested to modify the title of the paper due to the lack of human studies in both T1D and T2D.
5. Figure 1 is missing explanation in the text when it first appears.
6. In the last paragraph of the first section, it is important to specify that the discussion is referring to data on in vitro studies.
7. In the conclusion, limitations on the conclusions drawn (particularly in the first half of the conclusion) should be stated more strongly since the majority of data is based on animal models.
Author Response
The authors present a very well structured paper on the topic of gluten and diabetes risk. There are a few areas which would benefit from some clarifications. These have been outlined below.
1. If gluten entered our diet about 10,000 years ago and yet the annual increase in incidence of T1D has only been on the rise in the past few decades, it is unclear how gluten likely contributes to this rise in incidence. A bridge between this information and the mechanistic data discussed from animal models is required to help with the flow of the first part of the paper.
Thank you for the comment. This is a very good question and there may be several reasons why the incidence only began to rise during the past few decades. One of the reasons may be that modern wheat sources compared to ancestral wheat sources seem to be more diabetogenic, which is indicated by a study in NOD mice by Gorelick et al.
Gorelick J., Yarmolinsky L., Budovsky A., Khalfin B., Klein J.D., Pinchasov Y., Bushuev M.A., Rudchenko T., Ben-Shabat S. The Impact of Diet Wheat Source on the Onset of Type 1 Diabetes Mellitus-Lessons Learned from the Non-Obese Diabetic (NOD) Mouse Model. Nutrients. 2017;9:482 doi: 10.3390/nu9050482.
Other reasons may include exposure to other environmental factors than gluten during the last few decades such as pollutants etc.
We have now added possible explanations to this question in section 2. “Gluten and T1D”.
2. In the section 2.2, the conclusion seems a bit strong for the data presented. For example only 1/3 studies mentioned found an effect of a GF diet during pregnancy and risk of T1D in their children and reference 49 was not significant. It is important to be transparent, thus more descriptions of the studies should be added, including limitations of the evidence in order to promote an unbiased discussion. For example, regarding references 47 and 48, please describe if there was a control group. If not, the study may be confounded by post-diagnosis standard treatment. Also, please indicate what gluten is being replaced with in the GF diets as this can be relevant to any effect observed. The conclusion drawn in this section is also a principal statement in the abstract, thus may also require modification.
We are now citing more references demonstrating reduced incidence of a GF diet in NOD mice and we have added to the descriptions of the studies as you suggested. We have also described what gluten was replaced with (protein from meat, casein or egg white) in the GF diets.
3. In section 2.3, again the conclusion does not seem well supported by the evidence presented. There was only one study mentioned which was in NOD mice which included a GF intervention. This section is lacking any discussion of the effect of a GF diet.
We have added more studies demonstrating changes in intestinal microbiota from NOD mice on a GF diet. We have also toned down the conclusion a little. Moreover, besides the observed changes in the intestinal microbiota from a GF diet, other changes in the intestine like less enteropathy and inflammation have been demonstrated from a GF diet, which is described in subsequent sections (2.4 and 2.5).
4.There is a general lack of human studies on gluten and T1D. There is some very limited evidence presented on effects when consumed during pregnancy, but none other than this. Furthermore, the case-control study in section 2.6 demonstrated a lower incidence in those with an occupation with grain crops, i.e. bakers. This further adds to the debate on any effect of a GF diet on T1D risk. The lack of information in human studies in T1D in order to draw any strong conclusions should be made clearer in the abstract. It is also suggested to modify the title of the paper due to the lack of human studies in both T1D and T2D.
The conclusions in the abstract have been toned down, and the title has been modified accordingly.
5. Figure 1 is missing explanation in the text when it first appears.
Thank you. Explanations are given on page 4, 6 and 7.
6. In the last paragraph of the first section, it is important to specify that the discussion is referring to data on in vitro studies.
Has now been corrected.
7. In the conclusion, limitations on the conclusions drawn (particularly in the first half of the conclusion) should be stated more strongly since the majority of data is based on animal models.
Has now been corrected.
Reviewer 3 Report
Haupt-Jorgensen et al. present a very well written and extremely interesting review regarding the link between gluten and diabetes. I have only minor comments.
Title: I believe that "with" would be the correct particle to use instead of "by"
2. Is there any more recent study predicting the incidence or stating what is happening now around the world?
2.2 please explain what every acronym is for when you first mention it.
It would be interesting to state if ref. No. 49 had their diets matched per carbohydrate content, as increased insulin sensitivity etc could merely be due a change in cho/protein proportions.
2.3 as microbiome is known to be influenced by nearly all dietary habits, again it would be interesting to mention whether these studies use appropriate control diets that only differ in gluten content.
Table 1. it would be nice to have table 1 organized in in utero and postnatal effects
3.4 last sentence: Maybe better to write “the results need further confirmation”.
Author Response
Haupt-Jorgensen et al. present a very well written and extremely interesting review regarding the link between gluten and diabetes. I have only minor comments.
Title: I believe that "with" would be the correct particle to use instead of "by".
Thank you for the comment. Has now been corrected.
2. Is there any more recent study predicting the incidence or stating what is happening now around the world?
To our knowledge there are no newer studies on the prediction of the global incidence of diabetes. However, there are newer national studies, but it seems more appropriate to cite the big global studies.
2.2 please explain what every acronym is for when you first mention it.
DAISY, DiViD and NHANES have now been defined in the abbreviation list. BABYDIAB and BABYDIET have no definitions. NOD, BB etc. have already been defined in the abbreviation list.
It would be interesting to state if ref. No. 49 had their diets matched per carbohydrate content, as increased insulin sensitivity etc could merely be due a change in cho/protein proportions.
The first-degree relatives included in the study (ref. 49) were educated to comply with the specific diets and were provided with gluten-free products. Compliance to the diets was monitored. My understanding is that their diets were not matched for carbohydrate content. I agree with your rationale, but on the other hand the glycemic index of gluten-free and gluten-containing food is most likely similar:
The glycaemic index of a range of gluten-free foods. S. C. Packer, A. Dornhorst, G. S. Frost. Diabet Med. 2000 Sep; 17(9): 657–660.
To clarify, we have now specified that the diets were not matched for carbohydrate etc.
2.3 as microbiome is known to be influenced by nearly all dietary habits, again it would be interesting to mention whether these studies use appropriate control diets that only differ in gluten content.
We agree and have now described the gluten-free and gluten-containing diets used in the studies in more detail in section 2.2.
Table 1. it would be nice to have table 1 organized in in utero and postnatal effects.
Table 1 has now been organized accordingly.
3.4 last sentence: Maybe better to write “the results need further confirmation”.
Has now been corrected.
Reviewer 4 Report
It is a very thorough review work, well organized and carried out by authors with numerous publications in this field. Very didactic. The only point that is not clear to me is the title, since it could be interpreted that the studies have already been done in humans, and it does not respond to the conclusions in which it is indicated that human studies are necessary
Author Response
It is a very thorough review work, well organized and carried out by authors with numerous publications in this field. Very didactic. The only point that is not clear to me is the title, since it could be interpreted that the studies have already been done in humans, and it does not respond to the conclusions in which it is indicated that human studies are necessary.
Thank you for the comment. The title has been modified accordingly.